# Sending-or-Not-Sending Twin-Field Quantum Key Distribution with Light Source Monitoring

**DOI:** 10.3390/e22010036

**Published:** 2019-12-26

**Authors:** Yucheng Qiao, Ziyang Chen, Yichen Zhang, Bingjie Xu, Hong Guo

**Affiliations:** 1State Key Laboratory of Advanced Optical Communication, Systems and Networks, Department of Electronics, and Center for Quantum Information Technology, Peking University, Beijing 100871, China; glqyc251@pku.edu.cn (Y.Q.); chenziyang@pku.edu.cn (Z.C.); 2State Key Laboratory of Information Photonics and Optical Communications, Beijing University of Posts and Telecommunications, Beijing 100876, China; zhangyc@bupt.edu.cn; 3Science and Technology on Security Communication Laboratory, Institute of Southwestern Communication, Chengdu 610041, China; xbjpku@163.com

**Keywords:** quantum key distribution, twin-fields, sending-or-not-sending, light source monitoring

## Abstract

Twin-field quantum key distribution (TF-QKD) is proposed to achieve a remote key distribution with a maximum secure transmission distance up to over 500 km. Although the security of TF-QKD in its detection part is guaranteed, there are some remaining problems in the source part. The sending-or-not-sending (SNS) protocol is proposed to solve the security problem in the phase post-selection process; however, the light source is still assumed to be an ideal coherent state. This assumption is not satisfied in real-life QKD systems, leading to practical secure issues. In this paper, we discuss the condition that the photon number distribution (PND) of the source is unknown for the SNS protocol, demonstrate that the security analysis is still valid under a source with unknown PND, and show that with light source monitoring, the performance of the SNS protocol can remain almost unchanged.

## 1. Introduction

Quantum key distribution (QKD) provides a way for different communication parties to share a set of identical security keys [1,2]. The security of QKD is based on the laws of quantum physics and has been proved theoretically in different ways in the past decades. In addition to security analysis, people have also started to study the performance of QKD and to further improve the performance of real-life QKD systems by proposing new protocols [3,4,5,6,7].

Among all proposed protocols, measurement-device-independent (MDI) protocol is meaningful, since it can make up all the security loopholes in the detection part of QKD systems [8]. MDI protocol has developed rapidly in recent years, with many achievements obtained [9,10,11,12,13,14,15,16,17,18,19]. In theory, the application of the decoy state method [20] has been further studied, and more optimized parameter estimation results are obtained to achieve a better performance [9,10,11]. Besides, different MDI experiment schemes have been proposed [12] and several meaningful experiment achievements, including the field test [13] and the network experiment [14] based on the MDI protocol, have been carried out. Meanwhile, the continuous-variable MDI protocol has also been developed [16,17,18,19]. With the developing of experimental research, the MDI protocol achieved a transmission distance of 404 km in 2016, which was the farthest distance for standard QKD at that time [15].

The transmission distance of QKD has been further increased recently. Based on the MDI protocol, an extraordinary protocol called twin-field quantum key distribution (TF-QKD) was proposed by Lucamarini et al. in 2018, which can increase the transmission distance to over 550 km on standard optical fibers without quantum repeaters [21]. Because of its outstanding advantage in transmission distance, TF-QKD has recently been the subject of a lot of follow-up research [22,23,24,25,26,27,28,29]. Actually, in the original protocol, the phase randomization process will lead to secure issues since the information of random phase is announced to Eve in the post-selection process [30]. In further research, different types of modified TF-QKD protocols have been proposed to improve the incomplete security analysis [22,23,24,25,26]. Through classifying the signals [22], or adding an extra test mode [23], or using a pre-selected global phase [25,26], the security loophole of the post-selected part can be resolved. With the modified protocols, long-distance TF-QKD experiments have already been implemented [28,29]. Recently, an exciting experimental work has shown that the transmission distance for TF-QKD can reach over 500 km in practice [28], showing that TF-type protocols can significantly improve the transmission distance with the currently available technology.

Among those TF-type protocols, the sending-or-not-sending (SNS) protocol proposed by Wang et al. is an effective scheme to combine with the decoy state method to solve the phase randomization problem. In the SNS protocol, there is no post-selection process for the random phases of the signal bits, which are used to generate the final secret keys, thus the problem is avoided [22]. By optimizing the proportion of the sending and not-sending signals, the SNS protocol can still achieve a very long transmission distance.

A remaining problem with the SNS protocol is that the prepared quantum state in the source is assumed to be an ideal coherent state. Actually, the assumption could be broken since the prepared state will deviate from the ideal coherent state due to the non-ideality of the practical laser [31]. Moreover, since the light source structure in the SNS protocol is similar to the BB84 protocol, there will also be an untrusted source problem [32,33,34,35,36] in the source part, causing the photon number distribution (PND) of the light source to be unknown, and the prepared state to no longer be a coherent state.

In this paper, we provide further discussion of the SNS protocol under the unknown PND condition (UPC). By analyzing the form of the prepared state in Eve’s view, it is shown that the security analysis in the SNS protocol is still valid without the coherent state assumption, and the final secret key rate can be derived naturally. By applying a light source monitoring (LSM) method proposed previously [37,38], all relevant parameters can be estimated compactly, thus the secret key rate of the SNS protocol under UPC can be obtained. Moreover, it is indicated that the performance of the SNS protocol under UPC can almost keep the same ideal source condition through the numerical simulation.

The paper is organized as follows. In Section 2, we first analyze the security of the SNS protocol under UPC and give the calculation method of the secret key rate for the SNS protocol under UPC, then introduce an LSM method to obtain tight bounds of the parameters needed in calculating the secret key rate. In Section 3, we show the simulation results by applying the LSM method in the SNS protocol under UPC. Finally, we provide conclusion to our work in Section 4.

## 2. SNS Protocol with LSM

### 2.1. Security Analysis under UPC

In the SNS protocol [22,27], Alice (Bob) prepares a coherent state with an intensity $\mu_{A}$ ($\mu_{B}$), a random modulated phase $\delta_{A}$ ($\delta_{B}$) and a global phase $\gamma_{A}$ ($\gamma_{B}$). Only the states where Alice and Bob choose the same intensity ($\mu_{A}=\mu_{B}=\mu$) will be retained for discussion, thus the joint state sent by Alice and Bob is $|\sqrt{\mu }e^{i(\delta_{A}+\gamma_{A})}\rangle |\sqrt{\mu }e^{i(\delta_{B}+\gamma_{B})}\rangle$. In Eve’s view, it has the convex form
(1)$$\rho_{AB}=\sum_{k}p_{k}\left(\mu \right)|\psi_{k}\rangle \langle \psi_{k}|,$$
where $|\psi_{k}\rangle$ refers to a joint state with total photon numbers *k*, and $p_{k}\left(\mu \right)$ refers to the probability that the joint state contains *k* photons totally [22]. Specifically, the single-photon part of the state has the form of
(2)$$|\psi_{1}\rangle =\frac{1}{\sqrt{2}}{[e^{i(\delta_{B}+\gamma_{B})}|0\rangle }_{A}{|1\rangle }_{B}+e^{i(\delta_{A}+\gamma_{A})}{|1\rangle }_{A}{|0\rangle }_{B}].$$
$|\psi_{1}\rangle$ plays an important role during the whole security analysis of the SNS protocol [22]. In fact, the equivalent virtual protocols discussed in the security analysis focus on the single-photon part and the security analysis could remain valid when the phase difference between the two parts of the single state ${|1\rangle }_{A}{|0\rangle }_{B}$ and ${|0\rangle }_{A}{|1\rangle }_{B}$ remains to be
(3)$$\Delta \phi =\delta_{A}+\gamma_{A}-\delta_{B}-\gamma_{B}.$$

Under UPC, the quantum state prepared by Alice (Bob) is no longer an ideal coherent state $|\sqrt{\mu }e^{i(\delta_{A}+\gamma_{A})}\rangle$ ($|\sqrt{\mu }e^{i(\delta_{B}+\gamma_{B})}\rangle$), but a state with arbitrary PND, which can be written as $|\psi_{A}\rangle =\sum_{k}e^{ik(\delta_{A}+\gamma_{A})}\sqrt{P_{k,A}\left(\mu \right)}{|k\rangle }_{A}$ ($|\psi_{B}\rangle =\sum_{k}e^{ik(\delta_{B}+\gamma_{B})}\sqrt{P_{k,B}\left(\mu \right)}{|k\rangle }_{B}$) after the modulation process, where $P_{k,A}\left(\mu \right)\;\left(P_{k,B}\left(\mu \right)\right)$ is completely unknown.

As a result, with the analysis shown in Appendix A, the joint state sent by Alice and Bob can still have the convex form
(4)$$\rho_{AB}=\sum_{n}p_{n}\left(\mu \right)|\psi_{n}^{\prime }\rangle \langle \psi_{n}^{\prime }|$$
in Eve’s view, with the *n*-photon state
(5)$$|\psi_{n}^{\prime }\rangle =\frac{1}{\sqrt{p_{n}\left(\mu \right)}}\sum_{k=0}^{n}\sqrt{P_{k,A}\left(\mu \right)P_{n-k,B}\left(\mu \right)}e^{ik\Delta \phi }{|k\rangle }_{A}{|n-k\rangle }_{B},$$
and the probability of *n*-photon
(6)$$p_{n}\left(\mu \right)=\sum_{k=0}^{n}P_{k,A}\left(\mu \right)P_{n-k,B}\left(\mu \right).$$

Specifically, the single photon part of $\rho_{AB}$ under UPC can be written as
(7)$$|\psi_{1}^{\prime }\rangle =\frac{1}{\sqrt{2}}{[|0\rangle }_{A}{|1\rangle }_{B}+e^{i\Delta \phi }{|1\rangle }_{A}{|0\rangle }_{B}]$$
with a symmetric condition
(8)$$P_{k,A}\left(\mu \right)=P_{k,B}\left(\mu \right)=P_{k}\left(\mu \right),$$
which is also assumed in the original SNS protocol [22]. This result indicates that the original security analysis can be directly applied to UPC, since the phase difference between ${|1\rangle }_{A}{|0\rangle }_{B}$ and ${|0\rangle }_{A}{|1\rangle }_{B}$ is still Δϕ, that is, $|\psi_{1}^{\prime }\rangle$ only has a global phase difference with $|\psi_{1}\rangle$ in Equation (Equation 2).

#### Secret Key Rate

Unlike the original TF-QKD, the SNS protocol divides all the transmitted light pulses into two categories, which are called signal windows and decoy windows [20,22]. In the signal window (equivalent to Z-basis), Alice (Bob) randomly chooses to send or not send a signal pulse, and the data are used to generate keys. In the decoy window (equivalent to X-basis), Alice (Bob) sends decoy states with different intensities and the data are used to estimate the count rate and phase error rate of the signals’ single photon part. In addition, there is a post-selection process for the random phases $\delta_{A},\delta_{B}$ to ensure more accurate estimation results, which originates from TF-QKD.

The secret key rate of the SNS protocol has been given as [22,27]:(9)$$R=2\epsilon \left(1-\epsilon \right)P_{1}^{L}\left(\mu_{s}\right)s_{1}^{L}\left[1-H\left(e_{1}^{ph,U}\right)\right]-fS_{Z}H\left(E_{Z}\right),$$
in which ϵ is the probability that Alice (Bob) chooses to send out a signal pulse (it can be preset in the protocol), $H\left(x\right)=-x\log_{2}x-\left(1-x\right)\log_{2}\left(1-x\right)$ is the binary Shannon entropy function, $P_{1}^{L}\left(\mu_{s}\right)$ is the lower bound of the probability that a state with intensity $\mu_{s}$ sent in signal windows contains single photon, $s_{1}^{L}$ and $e_{1}^{U}$ refer to the lower bound of the count rate and the upper bound of phase error rate for the single photon part of signals, $S_{Z}$ and $E_{Z}$ refer to the count rate and the bit error rate of the signals. In decoy windows, a three-intensity decoy method with intensities $\mu_{d_{0}}=0,\mu_{d_{1}},\mu_{d_{2}}$ and $\mu_{d_{2}}>\mu_{d_{1}}>0$ is used to estimate $s_{1}^{L}$ and $e_{1}^{U}$ and the results are obtained as [22]
(10)$$s_{1}^{L}=\frac{p_{2}\left(\mu_{d_{2}}\right)\left[S_{\mu_{d_{1}}}-p_{0}\left(\mu_{d_{1}}\right)S_{0}\right]-p_{2}\left(\mu_{d_{1}}\right)\left[S_{\mu_{d_{2}}}-p_{0}\left(\mu_{d_{2}}\right)S_{0}\right]}{p_{2}\left(\mu_{d_{2}}\right)p_{1}\left(\mu_{d_{1}}\right)-p_{2}\left(\mu_{d_{1}}\right)p_{1}\left(\mu_{d_{2}}\right)},$$
(11)$$e_{1}^{ph,U}=\frac{S_{\mu_{d_{1}}}E_{\mu_{d_{1}}}-p_{0}\left(\mu_{d_{1}}\right)S_{0}/2}{p_{1}\left(\mu_{d_{1}}\right)s_{1}^{L}},$$
where $p_{0}\left(\mu_{d_{k}}\right),p_{1}\left(\mu_{d_{k}}\right),p_{2}\left(\mu_{d_{k}}\right)\;\left(k=0,1,2\right)$ are the probabilities that a state with intensity $\mu_{d_{k}}$ sent in decoy windows contains zero, single or two photons, $S_{\mu_{d_{k}}},E_{\mu_{d_{k}}}$ are the count rate and bit error rate of a state with intensity $\mu_{d_{k}}$ sent in decoy windows.

In the original SNS protocol, the light source is assumed to be able to prepare ideal coherent states [22], thus the PND of the light source satisfies Poisson distribution for each of Alice and Bob, that is,
(12)$$P_{n,A}\left(\mu \right)=P_{n,B}\left(\mu \right)=P_{n}\left(\mu \right)=e^{-\mu }\frac{\mu^{n}}{n!}$$
for $\mu =\{\mu_{s},\mu_{d_{1}},\mu_{d_{2}}\}$, hence the probabilities $p_{n}\left(\mu \right)\;\left(n=0,1,2\right)$ obtained in Equation (Equation 6) can be directly calculated as:(13)$$\begin{array}{cc} \; & p_{0}\left(\mu \right)=P_{0}^{2}\left(\mu \right),p_{1}\left(\mu \right)=2P_{0}\left(\mu \right)P_{1}\left(\mu \right),\\ \; & p_{2}\left(\mu \right)=2P_{0}\left(\mu \right)P_{2}\left(\mu \right)+P_{1}^{2}\left(\mu \right). \end{array}$$
When $p_{n}\left(\mu \right)$ are known, the parameters $s_{1}^{L}$ and $e_{1}^{U}$ in Equations (Equation 10) and (Equation 11) can be estimated. In addition, the probability $P_{1}^{L}\left(\mu_{s}\right)$ is also obtained exactly from Equation (Equation 12). With these results, the secret key rate in Equation (Equation 9) can eventually be calculated.

However, as mentioned above, the probabilities $p_{n}\left(\mu \right)$ become unknown under UPC, though the security analysis can be held. Fortunately, as analyzed in Reference [39], the decoy-state method is still valid under UPC when the lower and upper bounds of $p_{n}\left(\mu \right)\;\left(n=0,1,2\right)$ are obtained, and the results of $s_{1}^{L}$ and $e_{1}^{U}$ can be rewritten as
(14)$$s_{1}^{L}=\frac{p_{2}^{L}\left(\mu_{d_{2}}\right)\left[S_{\mu_{d_{1}}}-p_{0}^{U}\left(\mu_{d_{1}}\right)S_{0}\right]-p_{2}^{U}\left(\mu_{d_{1}}\right)\left[S_{\mu_{d_{2}}}-p_{0}^{L}\left(\mu_{d_{2}}\right)S_{0}\right]}{p_{2}^{U}\left(\mu_{d_{2}}\right)p_{1}^{U}\left(\mu_{d_{1}}\right)-p_{2}^{L}\left(\mu_{d_{1}}\right)p_{1}^{L}\left(\mu_{d_{2}}\right)},$$
(15)$$e_{1}^{ph,U}=\frac{S_{\mu_{d_{1}}}E_{\mu_{d_{1}}}-p_{0}^{L}\left(\mu_{d_{1}}\right)S_{0}/2}{p_{1}^{L}\left(\mu_{d_{1}}\right)s_{1}^{L}},$$
where $p_{n}^{L(U)}\left(\mu \right)$ refers to the lower (upper) bound of $p_{n}\left(\mu \right)$. With Equation (Equation 6), $p_{n}^{L(U)}\left(\mu \right)\;\left(n=0,1,2\right)$ can be obtained as
(16)$$\begin{array}{cc} \; & p_{0}^{L(U)}\left(\mu \right)={\left[P_{0}^{L(U)}\left(\mu \right)\right]}^{2},p_{1}^{L(U)}\left(\mu \right)=2P_{0}^{L(U)}\left(\mu \right)P_{1}^{L(U)}\left(\mu \right),\\ \; & p_{2}^{L(U)}\left(\mu \right)=2P_{0}^{L(U)}\left(\mu \right)P_{2}^{L(U)}\left(\mu \right)+{\left[P_{1}^{L(U)}\left(\mu \right)\right]}^{2} \end{array}$$
with Equation (Equation 8), hence the secret key rate can still be calculated effectively if tight bounds of $P_{n}\left(\mu \right)$ are obtained.

### 2.2. Parameters Estimation with LSM

The source structure in the SNS protocol is similar to that of the BB84 and MDI protocol, therefore it is possible to apply the LSM scheme proposed in the BB84 and MDI protocol to the SNS protocol to estimate $P_{n}\left(\mu \right)$. Recently, a new LSM scheme proposed by us in the MDI protocol gives tight bounds of $P_{n}\left(\mu \right)$ in each of Alice’s and Bob’s part under UPC [38]. Since the SNS protocol and the MDI protocol have the same structure in the intensity modulation part, the same LSM module can be added to the SNS protocol as shown in Figure 1, and with the extra LSM module, the same results can be obtained in the SNS protocol as follows [38]:
(17)$$P_{0}^{U}\left(\mu \right)=P_{0}^{L}\left(\mu \right)=\frac{P^{\mu }\left(\eta_{0}\right)}{1-Y_{0}},$$
(18)$$P_{1}^{L}\left(\mu \right)=\frac{{\left(1-\eta_{2}\right)}^{2}P^{\mu }\left(\eta_{1}\right)-{\left(1-\eta_{1}\right)}^{2}P^{\mu }\left(\eta_{2}\right)}{\left(1-Y_{0}\right)\left(1-\eta_{1}\right)\left(1-\eta_{2}\right)\left(\eta_{1}-\eta_{2}\right)}-\left(\frac{1}{1-\eta_{1}}+\frac{1}{1-\eta_{2}}\right)P_{0}^{U}\left(\mu \right),$$
(19)$$\begin{array}{cc} P_{1}^{U}\left(\mu \right) & =\frac{\left(1-\eta_{2}\right)\left(1-\eta_{1}\right)}{\left[1-\eta_{2}-\left(1-\eta_{1}\right)\left(2-\eta_{1}\right)\right]}\{\frac{P^{\mu }\left(\eta_{1}\right)}{{\left(1-\eta_{1}\right)}^{2}\left(1-Y_{0}\right)}+\frac{\left[1-{\left(1-\eta_{2}\right)}^{3}\right]}{\eta_{2}{\left(1-\eta_{2}\right)}^{2}}P_{0}^{U}\left(\mu \right)\\ \; & +\frac{1-\eta_{2}}{\eta_{2}}-\frac{P^{\mu }\left(\eta_{2}\right)}{\eta_{2}{\left(1-\eta_{2}\right)}^{2}\left(1-Y_{0}\right)}-\frac{P_{0}^{L}\left(\mu \right)}{{\left(1-\eta_{1}\right)}^{2}}\}, \end{array}$$
(20)$$P_{2}^{L}\left(\mu \right)=\frac{P^{\mu }\left(\eta_{2}\right)}{\left(1-Y_{0}\right){\left(1-\eta_{2}\right)}^{2}\eta_{2}}-\frac{1-\eta_{2}}{\eta_{2}}-\frac{\left[1-{\left(1-\eta_{2}\right)}^{3}\right]}{{\left(1-\eta_{2}\right)}^{2}\eta_{2}}P_{0}^{U}\left(\mu \right)-\frac{2-\eta_{2}}{1-\eta_{2}}P_{1}^{U}\left(\mu \right),$$
(21)$$P_{2}^{U}\left(\mu \right)=\frac{P^{\mu }\left(\eta_{2}\right)}{\left(1-Y_{0}\right){\left(1-\eta_{2}\right)}^{2}}-\frac{P_{0}^{L}\left(\mu \right)}{{\left(1-\eta_{2}\right)}^{2}}-\frac{P_{1}^{L}\left(\mu \right)}{1-\eta_{2}}$$
with a condition
(22)$$\eta_{1}\left(2-\eta_{2}\right)>1,$$
where $\eta_{k}\left(k=0,1,2\right)$ is the variable attenuation coefficient in the extra LSM module, $P^{\mu }\left(\eta_{k}\right)$ is the probabilities of the single photon detector’s (SPD) not responding in the LSM module, and $Y_{0}$ is the dark count rate of the SPD. When $P_{n}\left(\mu \right)$ are estimated with the LSM scheme, the key parameters $s_{1}^{L},e_{1}^{U}$ in Equations (Equation 14) and (Equation 15) can be calculated, and the secret key rate of SNS protocol under UPC is obtained.

## 3. Performance with Numerical Simulation

The performance of the original SNS protocol and the SNS protocol with the LSM scheme can be compared with a determined light source condition. Considering an ideal simulation circumstance first [22], the PND of the light source, $P_{n}\left(\mu \right)$, is assumed to satisfy Equation (Equation 12). In this case, the probabilities $P^{\mu }\left(\eta \right)$ in the LSM scheme can be simulated as [38]
(23)$$P^{\mu }\left(\eta \right)=\left(1-Y_{0}\right)e^{-\eta \mu },$$
and the estimation results of $P_{n}\left(\mu \right)$ can be calculated with Equations (Equation 17)–(Equation 21). Except $P^{\mu }\left(\eta \right)$, other parameters used in Equations (Equation 9)–(Equation 11), (Equation 14) and (Equation 15), such as $S_{\mu_{d_{k}}},E_{\mu_{d_{k}}},S_{Z},E_{Z}$, can be simulated with the same method discussed in the original SNS protocol [22,27], hence the final secret key rate can be calculated.

To compare the performance of the LSM scheme and the original SNS protocol under ideal simulation conditions, the simulation parameters, which are listed in Table 1, are set to be the same as in Reference [22]. Besides, for the LSM scheme, we set the attenuation coefficient as $\eta_{0}=1,$
$\eta_{1}=0.95,$
$\eta_{2}=0.9$ to obtain tight estimation results of $P_{n}\left(\mu \right)$; other parameters, including the intensities $\mu_{s},\mu_{d_{1}},\mu_{d_{2}}$ and the sending probability ϵ, are optimized at different transmission distances for both the original protocol and the SNS protocol with the LSM scheme.

The simulation results shown in Figure 2 indicate that, with the LSM scheme, the SNS protocol under UPC can have a performance that is almost the same as that of the original protocol with an ideal source. Specifically, both cases have a maximum transmission distance of up to over 800 km, and the difference between the LSM scheme and the ideal source condition is only about 0.5 km. Besides, the difference of the secret key rate between them is only about 1% for a typical distance of 100 km.

In order to further analyze the practical performance of the SNS protocol and it with the LSM scheme, it is necessary to consider a more practical simulation circumstance. For a real-life QKD system, the PND of the source’s signals is not fixed although without Eve’s disturbance [31,40,41]. Under this condition, the performance of the original SNS protocol will be degraded even though the security problem of an unknown PND is ignored. Specifically, the signal sent out from the source can be considered as a fluctuated coherent state with a Gaussian-distributed average photon number μ, which has a probability distribution of
(24)$$P\left(\mu \right)=\frac{1}{\sqrt{2\pi }\sigma_{\mu }}\exp\left[-\frac{{\left(\mu -\mu_{0}\right)}^{2}}{2\sigma_{\mu }^{2}}\right],$$
where $\mu_{0},\sigma_{\mu }$ are the mean value and the standard deviation of μ. Without the LSM module, the probabilities $P_{n}\left(\mu \right)$ can only be estimated by assuming that μ belongs to a confidence interval, that is, $\mu \in [\mu_{L},\mu_{U}]$, with a confidence level $\varepsilon =\int_{\mu_{L}}^{\mu_{U}}P\left(\mu \right)d\mu$, where
(25)$$\mu_{L}=\left(1-\delta \right)\mu_{0},\mu_{U}=\left(1+\delta \right)\mu_{0},$$
hence the results are
(26)$$\begin{array}{cc} \; & P_{0}^{L(U)}\left(\mu \right)=e^{-\mu_{U(L)}},P_{1}^{L(U)}\left(\mu \right)=\mu_{L(U)}e^{-\mu_{L(U)}},\\ \; & P_{2}^{L(U)}\left(\mu \right)=\frac{\mu_{L(U)}^{2}}{2}e^{-\mu_{L(U)}}. \end{array}$$
For the LSM scheme, the probabilities $P^{\mu }\left(\eta \right)$ can be simulated as
(27)$$P^{\mu }\left(\eta_{i}\right)=\left(1-Y_{0}\right)\exp\left[-\eta_{i}\mu_{0}-\frac{{\left(\eta_{i}\sigma_{\mu }\right)}^{2}}{2}\right]$$
after considering the source fluctuation [36,37], and $P_{n}\left(\mu \right)$ can be estimated by $P^{\mu }\left(\eta_{i}\right)$ with Equations (Equation 17)–(Equation 21).

The performance of both the original SNS protocol and the original SNS protocol with LSM are simulated under different fluctuation coefficient $\sigma =\sigma_{\mu }/\mu_{0}$. The parameters are changed to be consistent with those in Reference [21] in Table 2 to simulate a more realistic condition, which is close to the practical experimental conditions [28], and $\epsilon ,\mu_{s},\mu_{d_{1}},\mu_{d_{2}}$ are optimized as well. For the original SNS protocol, the confidence levels are set as $\varepsilon =1-10^{-10}$, and in the LSM scheme, the attenuation coefficients are set as $\eta_{0}=1,\eta_{1}=0.95,\eta_{2}=0.9$. The simulation results under the practical simulation condition shown in Figure 3 indicate that the LSM scheme still performs well in practice, as its performance remains almost the same under the fluctuated light source. As a comparison, the performance of the original SNS protocol decreases obviously with a light source fluctuation σ=1%, and its transmission distance even reduces to less than 500 km when σ=2%. Since the condition σ>1% is common in real-life QKD systems [40,41], the LSM scheme will have a better performance compared to the original SNS protocol in practice, although only the fluctuation problem is considered.

## 4. Conclusions

In summary, we analyze the security of the SNS protocol under UPC, and propose an LSM scheme to solve the unknown PND problem in the source part. An important problem for the SNS protocol is whether its security analysis is still valid without assuming the prepared state is an ideal coherent state. In this paper, we calculate the form of the quantum state sent from Alice and Bob in Eve’s view, and show that the single photon part of the state has the same form as the ideal source condition under UPC, thus the security analysis remains valid. We apply the LSM scheme proposed previously to the SNS protocol to estimate the probabilities $p_{n}\left(\mu \right)$ precisely to eventually obtain a tight bound of the secret key rate. Through simulation, we show that, with the proposed LSM scheme, the performance of the SNS protocol under UPC can be almost the same as that of the original SNS protocol with an ideal source. Moreover, the LSM scheme improves the performance of the SNS protocol when considering source fluctuation, indicating that the SNS protocol can still have a long transmission distance with a fluctuated source in real-life QKD systems.

## Figures and Tables

**Figure 1 entropy-22-00036-f001:**
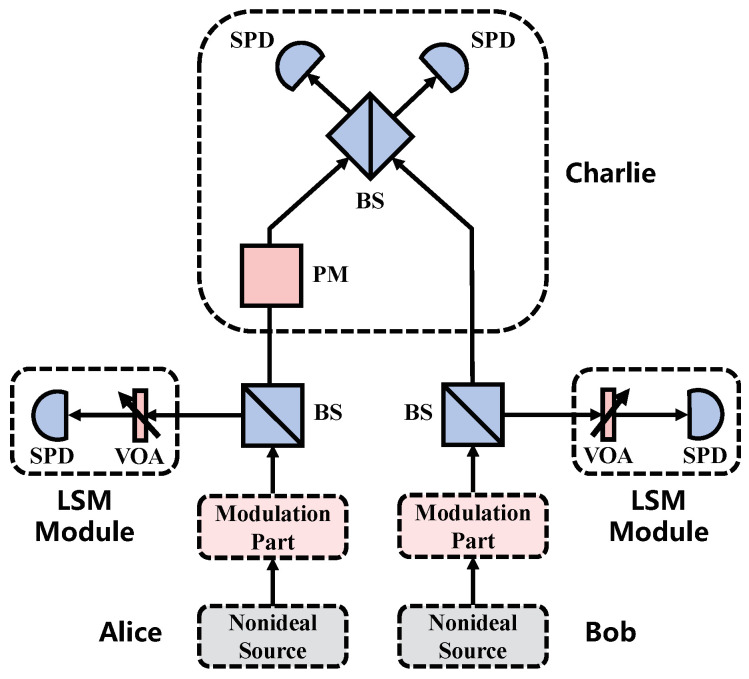
The structure of the sending-or-not-sending (SNS) protocol with an extra LSM module in each of Alice’s and Bob’s parts. The LSM module is made up of a variable optical attenuators (VOA) and a single photon detector (SPD). By changing the attenuation coefficient of the VOA, various sets of the results on the responding probability of the SPD are obtained, which can be used to estimate $P_{n}\left(\mu \right)$ effectively. The details of the monitoring scheme have been discussed in Reference [38].

**Figure 2 entropy-22-00036-f002:**
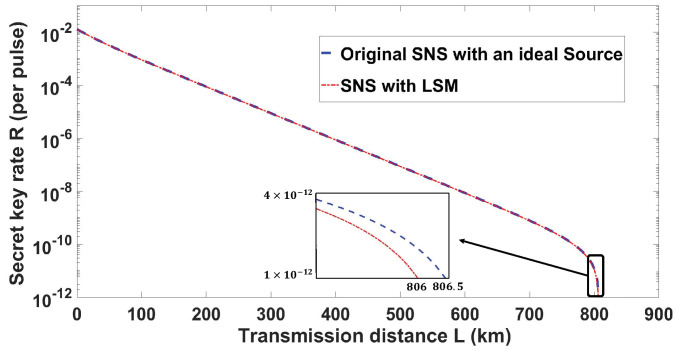
The performance of the proposed LSM scheme (red dash curve) compared to original SNS protocol (blue dash curve) with the parameters set as Table 1. The ratios of secret key rate between the LSM scheme and original SNS protocol are about 99.1%,98.9%,98.8% at the distance of 100, 300, 500 km, and the maximum transmission distances of the LSM scheme and original SNS protocol are about 806.0 and 806.5 km.

**Figure 3 entropy-22-00036-f003:**
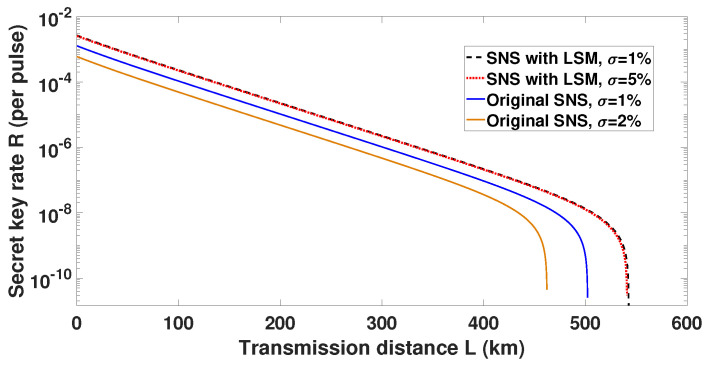
The performance of the LSM scheme with an untrusted and fluctuated light source compared to original SNS protocol. σ: the fluctuation coefficient. For the LSM scheme, we consider a small fluctuation condition σ=1% (black dash curve) and a large fluctuation condition σ=5% (red dash curve), and the performance between them is still close. For the original protocol, we consider a small fluctuation condition σ=1% (blue curve) and a relatively large fluctuation condition σ=2% (yellow curve), since the condition σ=2% already has an obviously worse performance than σ=1%.

**Table 1 entropy-22-00036-t001:** Values of parameters used in simulation. α: the fiber loss coefficient (unit: dB/km); $Y_{0}$: the dark count rate of the detector; $\eta_{D}$: the detection efficiency; $e_{\det}$: the misalignment error of the QKD system; *f*: the error correction efficiency.

α	$Y_{0}$	$\eta_{D}$	$e_{\det}$	*f*
0.2	$1.0\times 10^{-11}$	80%	1%	1.1

**Table 2 entropy-22-00036-t002:** Values of parameters used in simulation (set as in Reference [21] for a more practical condition).

α	$Y_{0}$	$\eta_{D}$	$e_{\det}$	*f*
0.2	$1.0\times 10^{-9}$	30%	3%	1.15

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
