# Peer review of "Sending-or-Not-Sending Twin-Field Quantum Key Distribution with Light Source Monitoring"

_entropy, 2019, doi:10.3390/e22010036_

Round 1

Reviewer 1 Report

An interesting paper on a relevant topic. Quantum cryptography is now in trend. The experimental data at a distance of 500 km is very promising. The proposed protocol is described well, but there is a lack of generalized experimental data under various environmental conditions. I recommend that you familiarize yourself with European literature in more detail.

Author Response

[Response to the comments]:

Thanks you very much for pointing out the lack of experimental data in our manuscript. Based on your comments, we have added a brilliant experimental work in Europe as a reference (Ref. [28]), as well as a meaningful theoretical work done in Europe (Ref. [26]). We have quoted the experimental work[28,29] to show that TF-type QKD has already been implemented practically. The results of Ref. [28] show that tolerant loss can reach over 90 dB in practical experimental conditions, and the transmission distance can exceed 500 km (by using ultralow-loss fiber). At the same time, relevant experimental work in China has also achieved a distance of 300 km with TF-QKD (Ref. [29]). The data and results in different experiments indicate that TF-QKD still has the advantage in long-distance QKD under practical environmental conditions.

On the other hand, as a numerical work, we tried to simulate different enviroments condition to analyze the performance of our monitoring scheme. We first set the experimental parameters to very ideal values in Table 1, which are the same as Ref. [22], to compare with the original SNS TF-QKD. In Table 2 we changed to use experimental parameters set as Ref. [21], which is close to the practical experimental conditions in Red. [28]. We have added a description of selection for the experimental parameters, hoping to show that our proposed monitoring scheme has superior performance under both ideal or practical conditions.

In general, the revisions to the review comments are listed below:

We have added 2 more references, Ref. [26] and Ref. [28]. We have added several sentences in line 33-43 to introduce TF-QKD and related experimental results for details. We have added a sentence in line 111-113, and a sentence in line 123-125 to explain the choice of experimental parameters in the simulation.

These changes are marked in red in our revised manuscript, and you can see the attached pdf file for details.

Thank you again for your comments and suggestions!

[About the English language and style]:

We notice that you choose “Extensive editing of English language and style required.” According to your requirement, we have tried our best to improve our manuscript.

Changes in English style and grammar are highlighted in yellow in our revised manuscript, and the reviewerer can see the attached pdf file for details. We hope that the corrections will meet with the reviewer’s approval.

Reviewer 2 Report

In this mathematical and numerical work, the authors study theoretically the security of sending-or-not-sending protocols. The authors propose a scheme to combat the photon number distribution problem and show the performance can be improved to obtain significant transmission distances.

I recommend that the authors make minor changes to their nice work.

I have the following comments:

1) The introduction needs to be fleshed out considerably. At the moment lists of works are given, but not detailed explanations of the importance and physics of the references, which is required in good introductions and to attract the attention o

2) Equation (19) needs to be broken up over 2 lines.

3) H is not defined in Eq. (9)

Author Response

[Response to the comments]:

Thank you very much for your comments. According to your 3 comments, we revised our manuscript as below:

1. We have enriched our introduction part, added a more detailed introduction to the work in the references. We have extended the introduction to the references both for the MDI protocol and the TF protocol. For MDI-QKD, we have added introductions of theoretical work that combine the decoy state scheme, which is relevant to our work; and have added introductions of several meaningful experimental achievements, including the field test, the network experiment, and the experiment with the longest transmission distance for MDI. For TF-QKD, we have added 2 more references, and have added introductions to several theoretical work that resolved security issues for original TF protocol with different methods. We finally added introductions of a recent experimental work which can realized 500km communication with TF-QKD (combining the comments of another reviewer).

2. We thank the reviewer for pointing out a formatting issue. We have corrected the format of Eq. (19).

3. We thank the reviewer for pointing out that the parameter explanation in the formula is not clear enough. We have added an explanation of the function H, which is the binary Shannon entropy function, in the manuscript.

In general, the revisions to the review comments are listed below:

We have added several sentences in line 20-30 and line 33-43 to extende the introduction part, introduced more about the relevant work of MDI-QKD and TF-QKD. We have changes Eq. (19) into 2 lines. We have added a sentence to explain the definition of function H under Eq. (9).

These changes are marked in red in our revised manuscript, and the reviewerer can see the attached pdf file for details. We hope that the revision can be accepted by the reviewer.

Thank you again for your comments and suggestions!

[About the English language and style]:

We noticed that the review chooses “English language and style are fine/minor spell check required.” We have tried our best to further improve our manuscript.

Changes in English style and grammar are highlighted in yellow in our revised manuscript, and the reviewer can see the attached pdf file for details. We hope that the corrections will meet with the reviewer’s approval.
